# The Potential Environmental Impact of PAHs on Soil and Water Resources in Air Deposited Coal Refuse Sites in Niangziguan Karst Catchment, Northern China

**DOI:** 10.3390/ijerph16081368

**Published:** 2019-04-16

**Authors:** Chengcheng Li, Xin Zhang, Xubo Gao, Shihua Qi, Yanxin Wang

**Affiliations:** State Key Laboratory of Biogeology and Environmental Geology and School of Environmental Studies, China University of Geosciences, 430074 Wuhan, China; chengcheng009019@hotmail.com (C.L.); xinzhangcug@163.com (X.Z.); shihuaqi@cug.edu.cn (S.Q.); yx.wang@cug.edu.cn (Y.W.)

**Keywords:** polycyclic aromatic hydrocarbon, coal spoils, water contamination, soil pollution, karst catchment

## Abstract

Long-term deposition of coal spoil piles may lead to serious pollution of soil and water resources in the dumping sites and surrounding areas. Karst aquifers are highly sensitive to environmental pollution. In this study, the occurrence and release/mobilization of polycyclic aromatic hydrocarbons (PAHs) in coal waste and coal spoils fire gas mineral (CSFGM) were evaluated by field and indoor investigations at Yangquan city, one of the major coal mining districts in the karst areas of northern China. Field investigations showed that dumping of coal waste over decades has resulted in soil and water pollution via spontaneous combustion and leaching of coal spoil piles. Indoor analysis revealed that the 2-ring and 3-ring PAHs contribute to 65–80% of the total PAHs in coal spoils, with naphthalene (Nap), Chrysene (Chr), and Phenanthrene (Phe) as the dominant compounds. Based on a heating/burning simulation experiment, the production of PAHs is temperature-dependent and mainly consists of low-ring PAHs: 2-ring, 3-ring, and part of the 4-ring PAHs. The PAHs in the leachate are light-PAHs (Nap, 20.06 ng/L; Phe, 4.76 ng/L) with few heavy-PAHs. The distribution modes of PAHs in two soil profiles suggest that the precipitation caused downward movement of PAHs and higher mobility of light-PAHs.

## 1. Introduction

Significant environmental pollution and health impact issues caused by coal using and mining activity have been reported widely [1,2,3,4,5,6,7,8]. To some extent, the environmental problems and health impact caused by coal spoil are similar to coal and coal using. Several environmental problems and health impacts have been found in the areas where air deposited coal gangue dump banks exist [9,10,11,12,13]. Toxic substances, including trace elements, hydrogen fluoride (HF), carbon monoxide (CO), sulphur dioxide (SO_2_), hydrogen sulfide (H_2_S), PAHs (Polycyclic Aromatic Hydrocarbons), and particulate matters are released into the atmosphere due to spontaneous combustion [14,15,16,17,18,19,20,21,22,23,24]. The environmental fate and behavior of PAHs have received wide attention owing to their potential toxicity and genotoxicity [25,26,27]. PAHs are abundant in the earth. There are two main sources of PAHs in the environment. Natural sources are related to the incomplete combustion of organic matter. PAHs can also be produced geologically when organic sediments are chemically transformed into fossil fuels such as oil and coal. Therefore, natural PAHs originate mostly from volcanic eruption, plant emissions, and fires. In a mining site and the surrounding areas, anthropogenic PAHs are raised from three sources (Figure 2). Spontaneous combustion is the primary source of PAHs [28,29,30,31,32,33,34]. Additionally, PAHs can result from the transportation of coal particles some distance from the piles through wind and water pathways. In general, the lack of vegetation and the general disturbed nature of the landscape may facilitate the dispersal of coal particles over the greater landscape. The third source could be from infiltration into the soils [35]. Some not-well-compacted soils (e.g., coarse sand) would promote the downward movement of water and particles deeper into the soil. Cancer is a primary human health risk of exposure to PAHs. PAHs have been linked to skin, lung, bladder, liver, and stomach cancers in well-established animal model studies [36]. Additionally, exposure to PAHs can cause cardiovascular disease and poor fetal development [37,38]. Therefore, research on the distribution of PAHs in coal spoils and soils around the depositing sites can be helpful in understanding the potential environmental impact.

Coal use has brought about massive coal spoil piles in China. As one of the largest coal production countries in the world, China had a production capability of 5.3 billion tons in 2017, which was five times more than that in 2000 (0.99 billion tons) [39]. The rapid growth of the mining industry in China over the last decades has resulted in accumulation of large amounts of mining waste. There are more than 1500 coal waste dumps, and the total sum of solid waste is more than 5 gross tons (Gt) in China [40,41,42]. The amount of coal gangue dumped into the environment is increasing at a rate of 1.5 to 2.0 billion tons per year. In Shanxi Province, the province with the highest coal production in China [43], there are more than 300 coal waste piles, and the coal wastes generated in 2008 were more than 3.1 Gt, occupying a land area of 300 ha.

The aim of this study is to assess the potential release of PAHs from coal spoil banks, under spontaneous combustion or a leaching process. The No. 2 coal mine in Yangquan city, Shanxi Province, northern China, was chosen for this study, where huge coal spoil banks have piled up for the past 30 years. Coal spoil and soil samples were collected in the field. Indoor heating and burning experiments were employed to study the release characteristics of PAHs during spontaneous combustion. The mobility deference of PAHs under leaching was evaluated by the investigation of the distribution mode of individual PAHs in underground soil profiles.

## 2. Geological Setting

The Yangquan coal mining district is located south-west of Yangquan city (Shanxi province, E 113°36′ N 37°53′). Until the end of 2004, the total known reserve of coal at Yangquan had been estimated to be 104 Gt, covering an area of 1051 km^2^. The major periods of coal formation are Carboniferous (Taiyuan Formation) and Early Permian (Shanxi Formation) (Figure 1) [44]. The coal-bearing strata are made up of nineteen coal seams, with a total average depth of 175 m. The coal-bearing strata overlie Ordovician and Cambrian marine limestone and are overlain by the Xiashihezi Formation (Early Permian), the Shangshihezi Formation (Later Permian), and Triassic clastic units. The lithology of the Taiyuan Formation is mainly consisted of quartz sandstone, siltite, silty mudstone, mudstone, limestone, and coal. The lithology of the Shanxi Formation is mainly composed by feldspar-quartz sandstone, siltite, silty mudstone, mudstone, and coal.

Health impact and environmental pollution were reported at Yangquan coal mine areas [45,46,47]. The trace elements (e.g., Hg, Pb, As, F) contained in leachate of gangue and organic pollutants (e.g., PAHs) in dust enhanced the risk of chronic morbidity and premature mortality, particularly from respiratory and cardiovascular diseases for humans. With a long-term exposure to PAHs, animals may suffer from an allergic skin response and breast tumors. Twenty-one coal spoil piles exist in the area with a total amount of 1.0 Gt coal wastes, produced as the by-product of coal mining by the total 194 sub-coal mines in the Yangquan coal mining district. Among them, over 1/3 was combusted or is suffering from spontaneous combustion. The coal spoil piles are mostly located in the upstream of the Yangquan city and Pingding County (Appendix A). Spontaneous combustion of them has resulted in serious atmospheric environmental pollution and health impact [16].

## 3. Materials and Methods

### 3.1. Sampling

Taking into consideration sampling locations, sampling time, sampling randomicity and uniformity, a total of 23 representative samples of coal spoils, coal spoils fire gas mineral (CSFGM), soil and leachate were collected from the hugest coal spoil bank No. 15 (coal mine No. 2, Appendix A and Figure 2), 1 km to the south of Yangquan. The coal spoil samples were collected according to their lithology: Limestone, shale, and sandstone. The CSFGM sample was collected from a coal spoils fire where it was covered with loess and sandy soil, and the CSFGM was separated from the cover materials by knife. Surface soil samples were collected in the front of the coal spoil banks with a distance of 5, 10, 50, 100, and 500 m (Figure 2, profile A-A’).

Two bore holes were dug until we reached the groundwater table for the collection of soil samples along the vertical profile. Seven soil samples were collected from the bore hole B1 with sampling depths of 0, 0.1, 0.2, 0.5, 1, 1.5, and 2 m, which reached one old weathered coal spoil layer. Five soil samples were collected from bore hole B2 until the groundwater table, with sampling depths of 0, 0.1, 0.2, 0.5, and 1 m (Figure 2). All samples were collected into a clean 1000 mL brown glass bottle (cleaned by EPA procedures) and stored at 4 °C pending transportation to the Lab for indoor testing. One leachate sample was collected at the front foot of the coal spoil bank, normally deposited longer than two years. The leachate sample was filtered through 0.45 μm membranes on site and then stored at 4 °C for indoor testing.

### 3.2. Experimental and Analytical Methods

The mineral compositions of the coal gangue, fired coal gangue, and CSFGM derived from spontaneous combustion were analyzed using X-ray diffraction (XRD, Bruker AXS D8-Focus X-ray diffraction, Karlsruhe, Germany). Although the lithology of coal spoil samples include limestone, shale, and sandstone, over 99% of the fired coal spoils belong to sandstone, according to our field investigation. Therefore, sandstone coal spoil samples were chosen for the indoor experiments to further study the release of PAHs during spontaneous combustion. Selected coal spoils and soil samples were dried in a vacuum freeze dryer and crushed to be less than 75 μm in particle size. The samples were extracted with dichloromethane in accordance with the U.S. EPA 3540C method with few modifications. About 20 g of the coal spoil sample and soil samples were Soxhlet-extracted (previously rinsed with redistilled dichloromethane, DCM) for 48 h with DCM. A known quantity of internal standard hexamethylbenzene was added to the samples prior to the instrumental analysis to check the recovery. PAHs were analyzed on a GC/MSD in our studies (6890N/5975 inert GC/MSD, Agilent, Wilmington, NC, USA). PAHs were separated on a DB-5 column (30 m, 0.25 mm i.d.). Helium was employed as the carrier gas at a flow rate of 1.5 mL/min. The injection temperature and detector temperature were 260 °C and 300 °C, respectively. The GC oven temperature was programed to: 60 °C for 5 min, 3 °C/min up to 290 °C, and hold for 40 min. There were 16 U.S. EPA preferentially controlled PAHs in most of the coal spoil samples and CSFGM sample. They were 2-ring PAHs Nap (naphthalene); 3-ring PAHs Acy (acenaphthylene), Ace (acenaphthene), Flu (fluorene), Phe (phenanthrene), and Ant (anthracene); 4-ring PAHs Fla (fluoranthene), Pyr (pyrene), BaA (benzo[a]anthracene), and Chr (chrysene); 5-ring PAHs BbF (benzo[b]fluoranthene), BkF (benzo[k]fluoranthene), BaP (benzo[a]pyrene), and DahA (dibenz[a,h]anthracene); 6-ring PAHs IP (indeno[1,2,3-cd]pyrene) and BghiP (benzo[ghi]perylene).

A simulated spontaneous combustion of coal spoils under an oxygen-deficient/oxygenic enriched situation was conducted in a horizontal tubular reactor. The heating test was employed to simulate the release under the oxygen deficit condition which normally appears deep inside the coal spoil banks. The burning test simulates the oxygen-enriched situation on the surface of the coal spoil piles. Previous studies indicated that there are three stages for the spontaneous combustion of coal spoils: Low-temperature oxidation (<400 °C), spontaneous heating (600 °C), and spontaneous combustion stage (>800 °C) [48]. In addition, the firing temperature for spontaneous combustion of coal spoils can be over 1000 °C [49,50]. Hence, in this study, coal spoil samples were heated/burnt in the sequence of 200 °C, 400 °C, 600 °C, 800 °C, and 1000 °C for four hours. The temperature increased at a rate of 20 °C/min with a gas pumping rate of 0.01 m^3^/min. In the heating test, nitrogen was used as the flow gas while oxygen was applied in the burning test. The PAHs produced were collected in all experiments by adsorption over XAD-2 resin placed at the outlet of the furnace throughout the whole experiments based on the U.S. EPA 0023A method and European Standard EN 1948-1. A set of resin trap was cleaned before being filled with XAD-2 resin. For the sampling of PAHs, recovery compounds were added to the resin prior to sampling. The analysis of PAHs was made following the same analytical methods as the samples obtained in the field. Heating and burning tests were repeated, and similar values were obtained.

Method blanks (solvent), three replicate analyses per sample, and spiked blanks (standards spiked into solvent) were analyzed. In addition, surrogate standards were added to each of the samples to monitor procedural performance and matrix effects. The recovery ratios for the surrogates in the samples (79.1–102.3%) fell within the reported ranges by US EPA [51,52]. The concentrations of PAHs were corrected for the recovery ratios for the surrogates. A known quantity of internal standard hexamethylbenzene was added to the samples prior to the instrumental analysis in order to quantify the PAH concentrations.

## 4. Results and Discussion

### 4.1. PAHs in the Coal Spoils

The minerals of the coal gangue samples were mainly kaolinite and quartz, with minor proportions of pyrite (Appendix A). The mineral assemblage of the fired gangue contained quartz, mullite, mascagnine, syngenite, lazurite, and anhydrite. The XRD detectable condensate compounds in the samples of CSFGM were salammoniac (NH_4_Cl), koktaite ((NH_4_)_2_Ca(SO_4_)_2_•H_2_O), and elemental sulfur.

The results for the individual PAH concentration of the coal spoils and CSFGM samples are shown in Table 1 and Figure 3. The total PAHs concentration (16 compounds) in coal spoils ranged between 500.02 ng/g and 706.87 ng/g (average concentration of 607.5 ng/g). It is worthwhile to note that these average amounts of total PAHs are approximately 1.8–91 times higher than those reported in case studies for coal gangue samples in Hong Kong (170 ng/g) [53], Serrinha (6.67 ng/g) [54], and Henan (347 ng/g) [55]. The dominant compounds in coal spoils were Nap (425.7 ng/g), followed by Chr (34.19 ng/g), Phe (25.51 ng/g), BaA (16.80 ng/g), BghiP (16.15 ng/g), BbF (14.40 ng/g), Ant (14.37 ng/g), BkF (13.50 ng/g), BaP (12.11 ng/g), Pyr (10.66 ng/g), Fla (6.94 ng/g), IP (6.70 ng/g), Acy (4.80 ng/g), Flu (4.71 ng/g), Ace (2.93 ng/g), and DahA (2.16 ng/g). The 2-ring and 3-ring PAHs contributed to about 65–80% of the total PAHs concentration in coal spoils while the contributions of 5-ring and 6-ring PAHs were lower than 3%. The far more exceeding amount of Nap than other compounds of PAHs is probably due to the type of coals [56,57]. Some kinds of coals (i.e., lignite A) are high in Nap. The PAHs in the CSFGM showed a similar occurrence pattern, with the 2-ring and 3-ring PAHs occupying 95.2% of the total PAHs and 1.15% for 5-ring and 6-ring PAHs. The total PAHs concentration in the CSFGM sample was 2088.5 ng/g, which is three to four times higher than that in coal spoils. The reason for such a high content of PAHs in the CSFGM sample may be that during the incomplete combustion of coal spoil, large amounts of PAHs are released into the atmosphere, thereby entering into CSFGM because of its hydrophobic nature [15]. It is noted that the major compounds of PAHs in CSFGM were Nap (1556.8 ng/g), Acy (167.6 ng/g), Ant (104.0 ng/g), Phe (84.60 ng/g), Flu (61.43 ng/g), and Fla (55.26 ng/g). This is due to the strong migration behaviors of these low-ring PAHs (especially Nap) considering their volatile and water-soluble characteristics. Concerning the toxic properties of PAHs, some countries such as Canada, Australia, and USA have published sediment quality guidelines, which are not available in China. As a consequence, the Canadian Council of Ministers of the Environment (CCME) sediment quality guideline was used here to estimate the potential of adverse effects resulting from PAHs pollution in Yangquan (Table 1). It is noticed that Nap in all samples far more exceeded the guideline. Additionally, other low-ring PAHs including Acy, Ace, Flu, Phe, and Ant for the CSFGM sample were above the threshold.

All the 2-ring, 3-ring, and part of the 4-ring PAHs (Fla and Pyr) were found in the leachate sample collected in the field with a total PAHs concentration of 40.72 ng/L (Table 2, Figure 3). The occurrence of individual PAHs in the leachate showed a similar pattern to that in coal spoils and CSFGM. Nap was identified as the highest concentration of individual PAHs in the leachate. 4-ring PAHs (BaA and Chr) and 5-ring and 6-ring PAHs (heavy PAHs) were under the detection limit in the leachate. Absence of heavy PAHs in the leachate sample and presence in coal spoil samples can be referred to the low mobilization ability of heavy PAHs under the leaching condition.

### 4.2. Distribution of PAHs in Surround Surface Soil

The soil samples were silt, silty loam, and sandy loam and were typically associated with kaolinite, quartz, and minor proportions of ferroactinolite, lazurite, and pyrite. PAHs in soils were investigated around the coal spoil bank. The complete data set for individual PAHs and ΣPAHs in the soil samples are given in Table 3. The total PAHs concentration in soil samples ranged from 47.13 ng/g to 705.3 ng/g with an average value of 259.96 ng/g. Over 80% of the total PAHs were contributed by the 2-ring, 3-ring, and 4-ring PAHs while the contents of 5-ring and 6-ring in most soil samples were low. The highest contents of PAHs found in the soil samples were Nap, followed by Phe, Fla, and Flu whilst IP, DahA, and BghiP were mostly lower than the detection limit. It is clear that the two to four-ring PAHs are the major pollutants in soils around the coal spoil deposit area. The potential reason for this is the high mobilization ability of these low molecular weight PAHs in soils. Absence of heavy PAHs in the leachate sample and presence in the coal spoil samples may be due to the low mobilization ability of heavy PAHs under the leaching condition.

Sample location appears to be the most important factor affecting soil PAHs contents according to the data set of Profile (S1–S5, Figure 4). It is interesting to note that the highest total and individual PAHs concentration are not found in soil sample S1, which is the nearest one from the coal spoil piles. In general, surface soils taken 10–50 m away from the coal spoil piles (such as soil samples S2, S3, and B1-1) have higher contents of PAHs (285.2 ng/g, 702.2 ng/g, and 624.0 ng/g, respectively). Although the total PAHs content in S1 soil is not as high as that in S2 and S3, it is still higher than that in S4 and S5. Along the A-A’ profile, a decreasing trend of PAHs concentration in soil is observed, predominantly due to the fact that total PAHs concentrations usually decrease with the increase of the closeness to a contamination source [58].

### 4.3. Vertical Distribution of PAHs in Soil

Distribution of PAHs in soils with depth was investigated by digging borehole B1 and B2 (Figure 5 and Figure 6). The highest concentration of total PAHs was observed in the soil sample with depth of 0.2 m, followed by the surface soil, and soil with depth of 0.1 m. Along the profile of borehole B1, the PAHs concentration in the top soils (soils with depth less than 0.2 m) was higher than that of the soils from the bottom (soils B1-6 and B1-7; Figure 5). For the individual PAHs (except for Acy), the highest concentration was found in the three top soil samples. In addition, the concentration of individual PAHs in the bottom soils (soils B1-6 and B1-7) was lower than that of the top soils. Close to the old coal spoil layer, the PAHs in the bottom soils with depth of 2 m may come from the weathering of old coal spoils; hence, the elevation of PAHs concentration in these two soil samples is acceptable. Low concentration of total PAHs and individual PAHs in the soils with depth of 0.5 and 1.0 m indicates that the downward movement of PAHs is slow and retarded. It is clear that coal spoils can lead to PAHs pollution of soils around the depositing area, and the pollution of PAHs normally happens in the top soils.

Along the vertical profile of borehole B2, the highest total PAHs concentration was observed in the soil sample with a depth of 0.5 m (Figure 6). Among the 16-U.S. EPA PAHs, IP, DahA, and BghiP were not detected in all the soils from borehole B2 (except for BghiP in B2-1). For light-PAHs (all of the 2-ring, 3-ring, and part of the 4-ring PAHs: Fla and Pyr), the highest concentration was found in the soils collected with a depth of 0.5 m (B2-4). For the rest 4-ring, 5-ring, and 6-ring PAHs (heavy PAHs), the highest concentration was observed in the surface soils (B2-1). Located in the downstream of the coal spoil piles, the soils from borehole B2 are frequently rinsed by the outlet from coal spoil banks. The elevated concentration in light PAHs in soils B2-4 (0.5 m) further confirms the downward movement and retardation of heavy PAHs in the underground system. Due to the precipitation recharge, soil in this depth is rinsed frequently. Based on the difference of adsorption ability of light and heavy PAHs in soil [59,60], the light PAHs are easier to move downward than the heavy PAHs. Therefore, the heavy PAHs are left and enriched in the soil. The difference of movement ability of individual PAHs is further proved by the test results of the leachate sample collected in the field (Table 2, Figure 3). The highest concentration of PAHs found in the leachate was Nap (20.06 ng/L), followed by Phe (4.76 ng/L), Ant (4.48 ng/L), Flu (3.94 ng/L), Acy (3.49 ng/L), and Ace (2.29 ng/L). This also informs us that the water is not the major medium for top soil PAHs pollution. Therefore, coal spoils constitute the major and even the only source of PAHs pollution in the coal spoil depositing area, and the study on the spontaneous combustion of coal spoils is necessary for better understanding the release way of PAHs.

### 4.4. PAHs Emissions During Spontaneous Combustion

The heating and burning test were performed to estimate the emission of PAHs from coal spoils. It is difficult to measure the emission in the field while simulation tests in the laboratory provide an effective way to estimate emissions of pollutants during spontaneous combustion [40]. The heating test is employed to simulate the release under the oxygen deficit condition while the burning test simulates the oxygen-enriched situation. The sandstone coal spoil sample with a total PAHs value of 650.7 ng/g was chosen for this test.

The release of PAHs at the heating test was investigated at several temperature grades (Appendix A; Figure 7). The total release of PAHs was 2373.2 ± 45 ng/g with the highest release of 606 ± 20 ng/g at 800 °C. The contribution of PAHs at a relatively lower temperature (200 °C and 400 °C) seems important in the heating test. Over 40% of PAHs release came from the lower temperature (200 °C and 400 °C) heating of coal spoils. About 90% of PAHs release happened when the heating temperature reached about 800 °C. Among the PAHs release, 2-ring PAHs occupied 86.4%, 3-ring 6.5%, 4-ring 4.0%, and 5-ring and 6-ring 3.1%. The highest individual PAHs release content was Nap, ranging between 198.5 and 538.0 ng/g with an average value of 409.9 ng/g. The second and third highest individual PAHs released were Phe and Pyr, with an average value of 18.6 and 6.87 ng/g, respectively. The lowest individual PAHs released was DaA, mostly lower than the detection limit.

The burning test produced a lower content of PAHs (Figure 8). The total PAHs released was 996.6 ± 30 ng/g. This release content was about half of that in the heating test. The highest production of PAHs was 399.0 ± 21 ng/g at 200 °C, while the lowest was 95.84 ± 9 ng/g at 600 °C. The PAHs release at a low temperature (200 °C and 400 °C) accounted for about 60% of the total in the burning test. This means that the release of PAHs at a low temperature is still remarkable. High temperature (from 800 °C to 1000 °C) burning can increase the production of PAHs [61,62]. The production of PAHs is an endothermic reaction. Elevated temperature may help the break, synthesis, and cyclization of the pyrolysis product which produces PAHs. The PAHs released mostly belong to low-ring PAHs, with 2-ring of 74.1%, 3-ring of 15.6%, 4-ring of 7.75%, 5-ring of 1.8% and 6-ring of 0.75%. Among the PAHs released, the highest individual was Nap, ranging between 36.06 and 343.5 ng/g with an average value of 146.9 ng/g. The second highest was Phe, with a range of 2.93 and 8.32 ng/g. The lowest individual PAHs released was DaA, which was lower than the detection limit in all tests.

Low-ring PAHs (2-ring, 3-ring, and part of the 4-ring PAHs) are the major production of coal spoil spontaneous combustion, according to the results of the heating and burning test. The total PAHs released in the heating test were about twice as those in the burning test. The release of PAHs is temperature dependent, and it is remarkable under the low temperature condition.

## 5. Conclusions

Spontaneous combustion of coal spoils has led to PAHs pollution at Yangquan coal mining areas. To further understand the mechanism of PAHs release, field and indoor experiments were performed with coal spoils from Yangquan city, Shanxi Province, one major coal mining district in northern China. Our investigation implied that, 1) dumping of coal waste over decades has resulted in PAHs pollution, mostly due to spontaneous combustion of the coal gangue bank; 2) low ring PAHs are the major products of spontaneous combustion of coal spoils. The total PAHs released in the heating test are about twice as those in the burning test; 3) an elevated total PAHs concentration in soil and leachate samples is observed around the coal gangue banks. The dominant PAHs were naphthalene (Nap), Chrysene (Chr), and Phenanthrene (Phe). Our study first timely and then quantitatively assessed the release of PAHs from coal spoils by indoor simulation test methods. This will be helpful for the defense and remediation of coal mining wastes in China.

## Figures and Tables

**Figure 1 ijerph-16-01368-f001:**
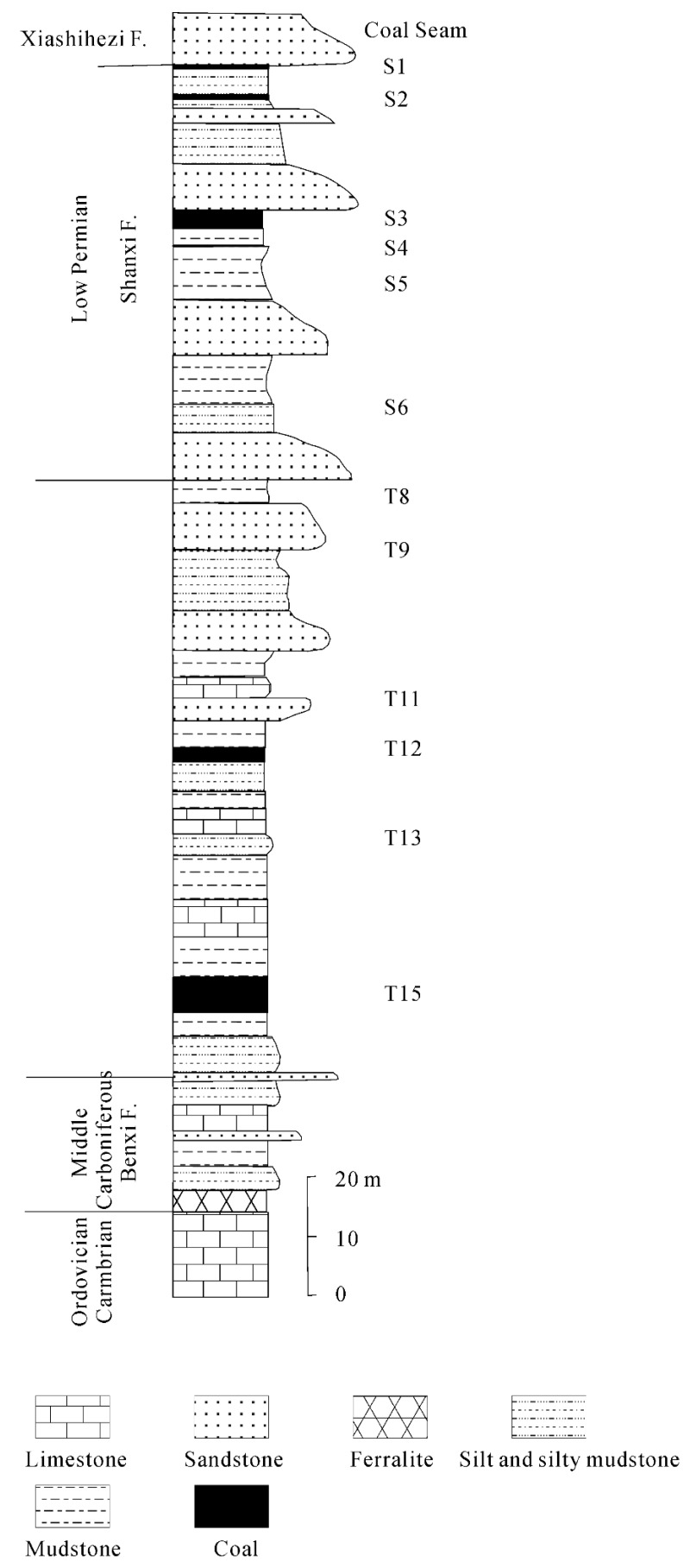
Stratigraphic series of the Carboniferous and Permian coal-bearing strata from Yangquan, Shanxi, China (Reproduced with permission from [Jiao & Wang], [Depositional environments of the coal-bearing strata and their controls on coal seams in the Yangquan mining district, Shanxi]; published by [Sedimentary Facies and Palaeogeography], 1999) [44].

**Figure 2 ijerph-16-01368-f002:**
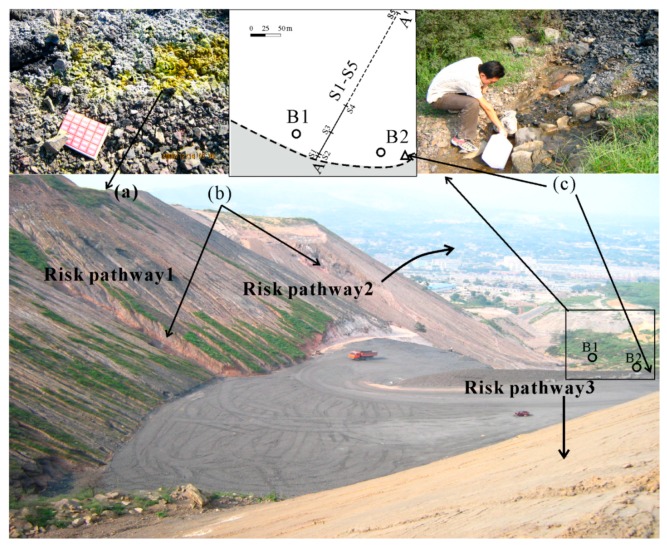
The largest coal spoil pile in Yangquan coal mine, No. 2 Coal Mine. (**a**) CSFGM, coal spoils fire gas minerals; (**b**) fired coal spoils; (**c**) leachate discharge from the coal spoil piles; S1–S5, soil samples collected in the front of the coal spoil bank with a distance of 5, 10, 50, 100, and 500 m (profile A–A’); B1 and B2, bore holes; bold dash line, the front boundary of the coal spoil bank. Also shown in the figure are the three risk pathways for polycyclic aromatic hydrocarbons (PAHs): Pathway 1, spontaneous combustion, and precipitation leaching of coal spoil piles; pathway 2, transportation of coal particles from the piles; pathway 3, infiltration of coal spoils into the soils.

**Figure 3 ijerph-16-01368-f003:**
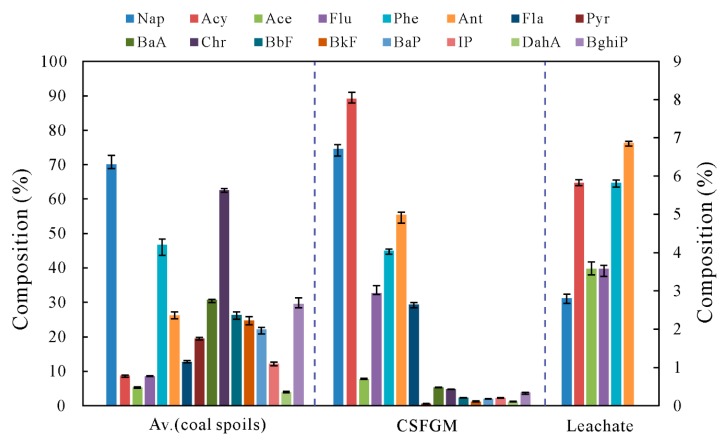
Distribution patterns of PAHs in coal spoils, CSFGM, and Leachate (Av.(coal spoils) is the average value of PAHs of the four coal spoil samples; Nap composition is shown in the left Y-axis, other PAHs composition is shown in the right Y-axis).

**Figure 4 ijerph-16-01368-f004:**
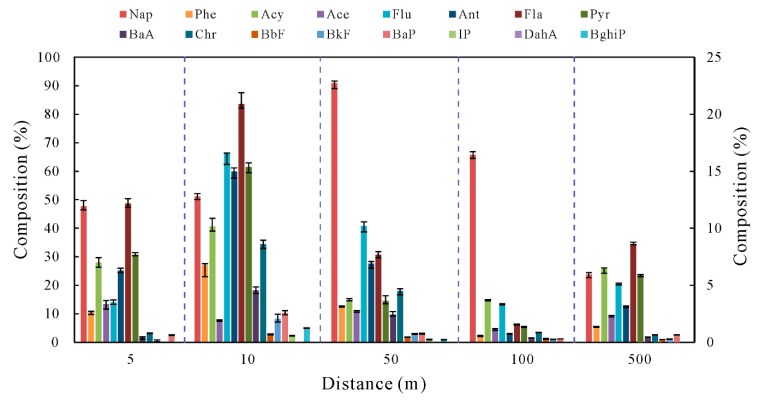
Distribution pattern of PAHs in soils around the coal spoil piles (Nap composition is shown in the left Y-axis, other PAHs composition is shown in the right Y-axis).

**Figure 5 ijerph-16-01368-f005:**
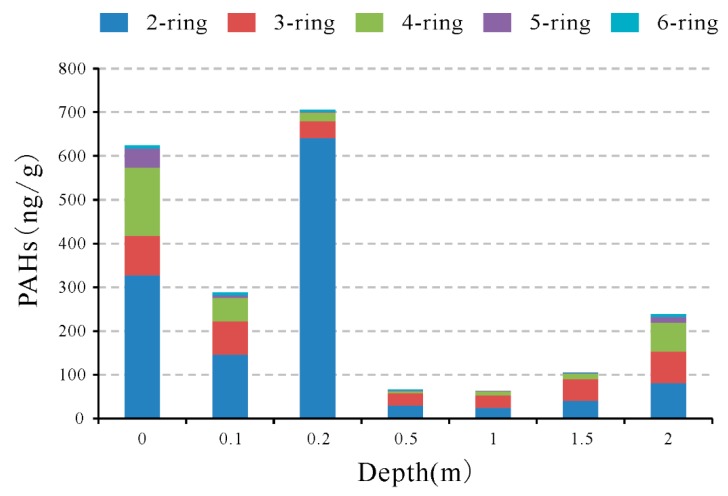
Concentration of PAHs vs. depth in the underground soils from borehole B1, according to ring number.

**Figure 6 ijerph-16-01368-f006:**
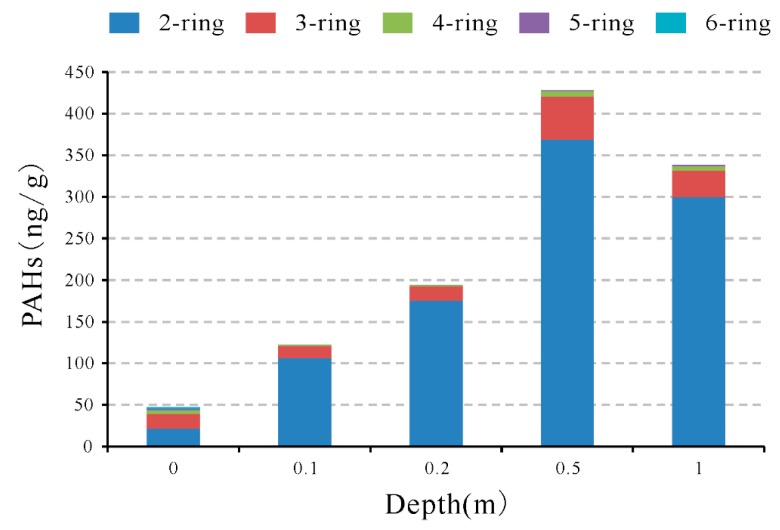
Concentration of PAHs vs. depth in the underground soils from borehole B2, according to ring number.

**Figure 7 ijerph-16-01368-f007:**
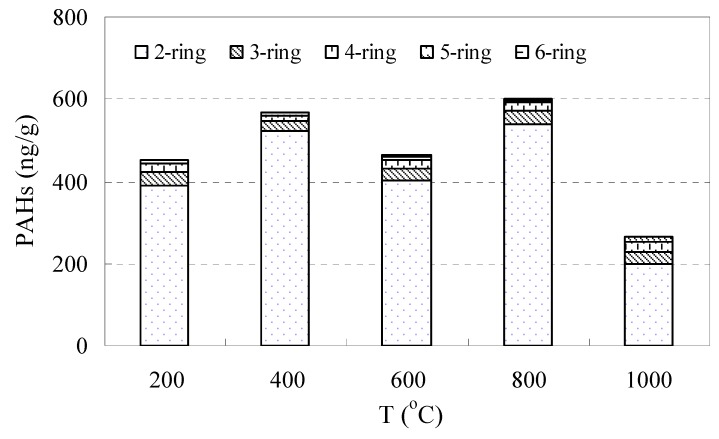
Release of PAHs in the oxygen-deficit heating test at different temperatures, according to ring number.

**Figure 8 ijerph-16-01368-f008:**
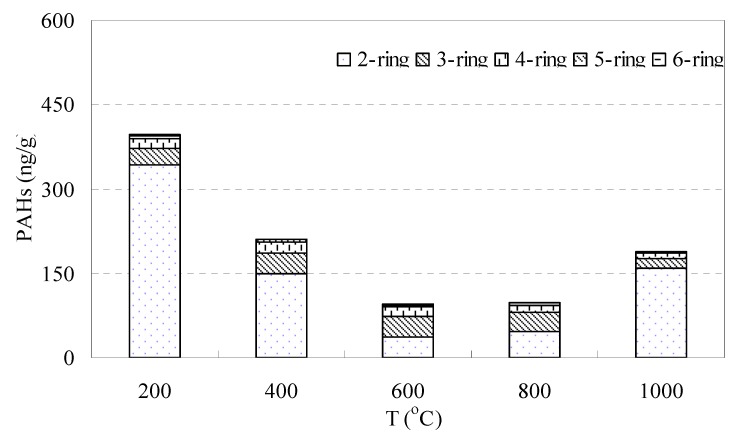
Release of PAHs in the oxygen-enriched burning test at different temperatures, according to ring number.

**Table 1 ijerph-16-01368-t001:** PAHs content in coal spoils, CSFGM (coal spoils fire gas mineral) sample, and their comparison with the Canadian Council of Ministers of the Environment (CCME) standard guideline (ng/g, dry weight).

ID	Nap	Acy	Ace	Flu	Phe	Ant	Fla	Pyr	BaA	Chr	BbF	BkF	BaP	IP	DahA	BghiP	∑PAHs
CG01	354.6	3.04	3.47	8.51	33.83	18.95	10.02	14.25	23.35	47.63	24.89	24.20	20.10	2.32	4.10	57.54	650.7
CG02	428.9	N.D.	N.D.	3.59	23.57	13.42	5.96	10.02	14.85	29.96	11.19	10.91	9.99	7.89	1.69	0.52	572.5
CG03	512.6	3.94	3.45	5.00	27.93	15.56	8.51	12.12	19.09	38.64	16.56	16.31	14.55	9.90	0.69	2.00	706.9
CG04	406.9	7.43	1.88	1.75	16.72	9.55	3.28	6.26	9.91	20.53	4.97	2.56	3.80	N.D.	N.D.	4.53	500.0
Av.	425.7	4.80	2.93	4.71	25.51	14.37	6.94	10.66	16.80	34.19	14.40	13.50	12.11	6.70	2.16	16.15	607.5
STDEV	65.75	2.32	0.91	2.86	7.21	3.94	2.96	3.41	5.76	11.62	8.44	9.11	6.91	3.93	1.75	27.64	90.42
CSFGM	1556.8	167.6	14.53	61.43	84.60	104.0	55.26	1.21	10.10	9.09	4.47	2.15	3.78	4.23	2.17	7.16	2088.5
CCME	34.6	5.87	6.71	21.2	41.9	46.9	111	53	31.7	57.1	0	0	31.9	0	6.22	0	0

Note: Av. is the average value of PAHs of the four coal spoil samples; STDEV is the standard deviation value of PAHs of the four coal spoil samples.

**Table 2 ijerph-16-01368-t002:** Chemical data and in situ physic-chemical parameters of the leachate sample collected in the field (concentration of ions in mg/L, PAHs in ng/L).

Parameters	Concentration	Parameters	Concentration	Parameters	Concentration	Parameters	Concentration
T (°C)	14.0	As	0.01	Nap	20.06	BkF	N.D.
pH	7.39	B	0.01	Acy	3.49	BaP	N.D.
EC	1252	Ba	0.05	Ace	2.29	IP	N.D.
Cl	21.3	Cr	0.07	Flu	3.94	DahA	N.D.
SO_4_	469.1	Cu	0.01	Phe	4.76	BghiP	N.D.
HCO_3_	194.1	Fe	0.01	Ant	4.48	∑PAHs	40.72
NO_3_	40.10	Mn	0.02	Fla	1.03		
Ca	141.2	S	149.3	Pyr	0.65		
K	0.50	Si	2.76	BaA	N.D.		
Mg	38.5	Sr	2.16	Chr	N.D.		
Na	91.8	Pb	0.01	BbF	N.D.		

Note: N.D., not detected.

**Table 3 ijerph-16-01368-t003:** PAHs content in soil samples collected around the coal spoil piles (ng/g, dry weight).

ID	Depth (m)	Nap	Acy	Ace	Flu	Phe	Ant	Fla	Pyr	BaA	Chr	BbF	BkF	BaP	IP	DahA	BghiP	∑PAHs
S1	0	51.28	7.50	3.58	3.73	11.21	6.76	13.05	8.24	0.47	0.86	0.04	0.01	0.68	0	0	0	107.4
S2	0	145.63	10.92	2.04	17.77	28.90	16.09	22.46	16.49	4.87	9.23	0.74	2.22	2.74	1.62	0	3.53	285.2
S3	0	638.91	3.96	2.87	10.94	13.27	7.41	8.25	3.88	2.59	4.80	0.47	0.78	0.81	1.69	0	1.62	702.2
S4	0	32.72	3.96	1.22	3.56	2.17	0.80	1.69	1.43	0.41	0.95	0.30	0.28	0.32	0	0	0	49.82
S5	0	13.11	6.95	2.45	5.49	5.83	3.37	9.26	6.33	0.48	0.70	0.25	0.30	0.69	0	0	0	55.22
Av1.	0	185.04	7.56	2.77	10.92	13.02	7.17	11.95	7.55	2.58	4.97	1.91	1.52	1.73	1.48	0.44	2.54	259.96
STDEV1	0	204.55	2.82	0.99	6.76	10.68	6.02	16.28	10.10	4.33	8.61	4.62	3.34	2.72	0.41	0.20	1.61	229.25
B1-1	0	327.89	9.21	2.82	23.07	34.85	19.38	64.31	39.64	17.80	35.12	18.15	13.64	11.16	1.10	0.51	5.30	624.0
B1-2	0.1	146.13	10.97	2.04	17.84	29.33	16.13	22.74	16.64	4.96	9.27	0.92	2.24	3.07	1.69	0	3.70	287.7
B1-3	0.2	641.09	3.98	2.87	10.98	13.47	7.43	8.35	3.91	2.64	4.82	0.59	0.79	0.91	1.75	0	1.69	705.3
B1-4	0.5	30.11	7.86	2.79	8.10	7.30	2.09	1.97	1.12	0.29	0.96	0.76	0.35	0.43	0.73	0.21	0.92	66.00
B1-5	1	24.27	11.88	1.18	4.87	6.74	4.05	5.31	1.93	0.52	1.21	0.31	0.14	0.76	0	0	0	63.17
B1-6	1.5	41.13	12.17	2.79	12.73	13.32	7.61	7.80	2.48	1.73	0.96	0.16	0.18	1.15	0	0	0.88	105.1
B1-7	2	80.64	7.33	4.03	18.09	27.61	15.40	31.75	17.56	5.33	12.03	4.55	2.56	4.67	1.81	0.59	4.01	238.0
Av2.	0.8	184.47	9.06	2.65	13.67	18.95	10.30	20.32	11.90	4.75	9.20	3.63	2.84	3.16	1.42	0.44	2.75	298.5
STDEV2	0.8	227.80	2.93	0.87	6.35	11.42	6.64	22.11	14.10	6.08	12.24	6.58	4.86	3.84	0.79	0.26	1.98	265.2
B2-1	0	22.05	5.12	1.98	3.76	4.19	2.26	0.22	2.18	0.67	1.68	0.75	0.36	0.68	0	0	1.24	47.13
B2-2	0.1	106.41	4.75	3.00	4.12	1.61	1.00	0.19	0.72	0.11	0.26	0	0.06	0.22	0	0	0	122.5
B2-3	0.2	175.31	6.97	3.01	5.94	0.82	0.29	0.60	0.60	0.14	0.39	0	0	0.32	0	0	0	194.4
B2-4	0.5	368.96	7.43	5.29	22.37	10.60	5.86	3.36	3.26	0.13	0.20	0.27	0.17	0.07	0	0	0	428.0
B2-5	1	300.02	0	3.16	12.26	10.07	6.02	1.91	1.90	0.64	1.10	0.34	0.22	0.66	0	0	0	338.3
Av3.	0.4	194.55	6.07	3.29	9.69	5.46	3.09	1.26	1.73	0.34	0.73	0.27	0.20	0.39	0	0	0.25	226.1
STDEV3	0.4	140.92	2.95	1.21	7.87	4.63	2.70	1.37	1.10	0.29	0.64	0.31	0.14	0.27	0	0	0.55	155.8

Note: Av1. is the average value of PAHs of the soil samples. STDEV1 is the standard deviation value of PAHs of the soil samples. Av2. is the average value of PAHs of bore hole B1 soil samples. STDEV2 is the standard deviation value of PAHs of bore hole B1 soil samples. Av3. is the average value of PAHs of bore hole B2 soil samples. STDEV3 is the standard deviation value of PAHs of bore hole B2 soil samples.

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
