# Peer review of "The Potential Environmental Impact of PAHs on Soil and Water Resources in Air Deposited Coal Refuse Sites in Niangziguan Karst Catchment, Northern China"

_ijerph, 2019, doi:10.3390/ijerph16081368_

Round 1

Reviewer 1 Report

The manuscript Potential environmental impact of PAHs on soil and 2 water resources in air deposited coal refuse sites in 3 Niangziguan karst catchment, northern China” by Li et al. discusses about the PAHs in leachate and soils in Northern China. Before I accept this manuscript for publication, I suggest a major revision.

Major comments

Compare results obtained in this study with others especially because authors mentioned that PAHs levels were high

Check for grammatical errors

The soil types and properties were not mentioned or classified

Abstract

Page 1 line 14: Insert “of the” before major coal … and change “shows” to “showed”

Page 1 line 19: replace “reveals” with “revealed”

Introduction

Page 1 line 33: spell out HF, CO, SO2,

Delete “outweighing PAHs from natural sources

Page 49-50: is this in China?

Page 2 line 48: Spell out Gt

Page 2 line 45: is there any recent information/data on the the sentence “As one of the largest coal 44 production country in the world, China has a production capability of 2.5 billion ton in 2007 which is 45 two times more than that in 2000 (0.99 billion ton)”

Page 2 line 51: there are more than 300 gobs of what?

Page 2 line 51: change to …..coal wastes generated in 2008 was more than 3.1 gt, occupying a land area …..

Page 2 line 54: Yangquan No. 2 coal mine, in China?, was chosen

Page 2 line 55: change to “where a huge coal spoils banks have piled up for the past 30 years”

Page 2 lines 61-70: refer to figure

Page 2 line 71: provide some form of reference to the phrase “ environment pollution and health ……coal mine areas. And what pollutants were known or reported to cause these health issues?

Page 2 line 72: insert “in the area” after …piles exists”

Page 2 lines72-73: where are these 194 located?

Page 2 line 72: in the sentence ..”total amount of 1.0 gt”….amount of what?

Describe Yangquan and how it is related or unrelated to the study site

Figure 1: which direction is north?

Page 3 lines 90-93: refer to figure

Page 3: Explain the purpose for collecting soil samples from around the water table

Experimental & analytical methods

Page 4 line 112: about 20 g of the Sample. Is this soil? Please indicate

What was the purpose of hexamethylbenzene? Was it to check instrument sensitivity or recovery?

Where the soil samples dried prior to PAH analysis? If so, please indicate

Page 5 line 137-138: the recovery ratios for the surrogate.... reported ranges. What were these ranges?

Results and discussion

Page 5 line 143: move to materials and methods

Page 5 lines 150 to 155: move to materials and methods

Provide standard deviations for all average concentrations

Table S1 and S2 are the main data, please include these in the main paper and not as supplement

What is the possible reason for this trend and also explain the potential reasons why nap was most abundant PAH “The major compounds of PAHs in CSFGM is Nap (1556.8 ng/g), Acy (167.6 ng/g), 166 Ant (104.0 ng/g), Phe (84.60 ng/g), Flu (61.43 ng/g) and Fla (55.26 ng/g). Most of the 4-ring, 5-ring and 167 6-ring PAHs have lower contents in CSFGM compared to those in coal spoils.”

Page 6 line 184: change “are” to “were”

Authors should explain why the low molecular weight PAHs were most abundant

Any reasons for these results? “Though the total PAHs content in S1 soil is not as high as S2 and S3, it is still higher 193 than S4 and S5. Along the A-A’ profile a decreasing trend of PAHs in soil is observed.”

Page 7 lines 199, 200 and 202: replace “is” with “was”

Page 7 line 206: provide distance for the bottom soil. Was it 2 m, 0.5 m or what?

Figures 5 & 6 have a very poor legibility

Page 7 line 218: replace “are” with “were”

Page 7 line 220: did authors mean PAHs or PHAs?

Page 8 lines 225-226: Is there any reference to support the sentence “Based on the difference of adsorption ability of light 225 and heavy PAHs on soil, the light PAHs is easier to move downward than heavy PAHs.”

Page 8 line 227: soil or top soil?

Page 8 lines 231-232: re-write the sentence “Therefore, as the major, even the only, 231 source of PAHs pollution in the coal spoils depositing area,”

Page 8 line 235: delete “test” after the heating

Page 8 line 236: move “in the field” to after the “emission rate”

Page 8 line 244: is 200 C considered low temperature? I also do not get the sentence because at high temperature combustion, the high molecular weight PAHs are abundant. Can authors explain this trend since lower molecular weight pahs contributed abundantly?

How different does heating or burning affect PAH production?

Indicate p values for all statistical significance

Include SD for average concentations

Page 9 line 278: change “production of” to “products of”

Page 9 line 274: replace “are” with “were”

Page 9 line 281: replace “compounds are” with “PAHs were”

Page 10 line 282: assesse?

Page 9 lines 264-265: the highest ……followed by phe. Please delete

Author Response

Dear Sir/Madam,

       Thank you so much for your kind review and valuable comments. The manuscript has been well prepared according to these comments. Great efforts were also made to correct the mistakes and improve the quality of the manuscript. The revisions in the manuscript are marked up highlighted in yellow color. Below is a point-by-point response.

Response to Reviewer 1 Comments

Open Review

English language and style

( ) Extensive editing of English language and style required 
(x) Moderate English changes required 
( ) English language and style are fine/minor spell check required 
( ) I don't feel qualified to judge about the English language and style 

Yes

Can be improved

Must be improved

Not applicable

Does the introduction provide sufficient background and   include all relevant references?

( )

(x)

( )

( )

Is the research design appropriate?

(x)

( )

( )

( )

Are the methods adequately described?

( )

(x)

( )

( )

Are the results clearly presented?

( )

(x)

( )

( )

Are the conclusions supported by the results?

(x)

( )

( )

( )

Comments and Suggestions for Authors

The manuscript Potential environmental impact of PAHs on soil and 2 water resources in air deposited coal refuse sites in 3 Niangziguan karst catchment, northern China” by Li et al. discusses about the PAHs in leachate and soils in Northern China. Before I accept this manuscript for publication, I suggest a major revision.

Thanks for your progressive suggestion. We have modified the paper and made great efforts to improve the quality of the manuscript. The revisions in the manuscript are marked up highlighted in yellow color.

Major comments

Compare results obtained in this study with others especially because authors mentioned that PAHs levels were high.

We have added the comparison of the PAHs concentrations in this study and in other places in the revised manuscript, see line 183-186 and 464-469.

Check for grammatical errors.

We have checked the paper carefully and revised all the grammatical errors in the revised manuscript.

The soil types and properties were not mentioned or classified.

The soil samples were composed of silt, silty loam, and loamy sand, and were typically associated with kaolinite, quartz, and minor proportions of ferroactinolite, lazurite and pyrite. We have added the description of the soil types and properties in the revised manuscript, see line 213-214.

Abstract

Page 1 line 14: Insert “of the” before major coal … and change “shows” to “showed”.

We have modified this description in the revised manuscript, see line 14-15.

Page 1 line 19: replace “reveals” with “revealed”

 We have replaced “reveals” with “revealed” in the revised manuscript, see line 17.

Introduction

Page 1 line 33: spell out HF, CO, SO2,

We have spelled them out in the revised manuscript, see line 32-33.

Delete “outweighing PAHs from natural sources

We have removed “outweighing PAHs from natural sources” in the revised manuscript, see line 37-38.

Page 49-50: is this in China?

Yes, Shanxi Province is located in northern China and it has the highest coal production.

Page 2 line 48: Spell out Gt

We have spelled out “Gt” in the revised manuscript, see line 56.

Page 2 line 45: is there any recent information/data on the sentence “As one of the largest coal 44 production country in the world, China has a production capability of 2.5 billion ton in 2007 which is 45 two times more than that in 2000 (0.99 billion ton)”

Thank you. We have renewed the data in the revised manuscript, see line 53-54, and 433-434. China has a coal production of 5.3 billion ton in 2017.

Page 2 line 51: there are more than 300 gobs of what?

There are more than 300 gobs of coal waste dumps. We have corrected this sentence in the revised manuscript, see line 59.

Page 2 line 51: change to …..coal wastes generated in 2008 was more than 3.1 gt, occupying a land area …..

We have modified our statement in the revised manuscript, see line 59-60.

Page 2 line 54: Yangquan No. 2 coal mine, in China?, was chosen

No. 2 coal mine in Yangquan city, Shanxi Province, northern China, was chosen for this study. We have modified this statement in the revised manuscript, see line 62-64.

Page 2 line 55: change to “where a huge coal spoils banks have piled up for the past 30 years”

Thanks for your invaluable suggestion. We have changed “where the hugest coal spoils banks have piled up since thirty years ago” to “where a huge coal spoils banks have piled up for the past 30 years” in the revised manuscript, see line 63-64.

Page 2 lines 61-70: refer to figure

Thanks for your kind reminder. We have referred the figure (Figure 1) to this statement in the revised manuscript, see line 71-72 and 443-444.

Page 2 line 71: provide some form of reference to the phrase “environment pollution and health ……coal mine areas. And what pollutants were known or reported to cause these health issues?

We have added related references to the sentence “Environmental pollution and health impact were reported at Yangquan coal mine areas” in the revised manuscript, see line 79, and 445-451.

The trace elements (e.g., Hg, Pb, As, F) contained in leachate of gangue and organic pollutants (e.g., PAHs) in dust enhance the risk of chronic morbidity and premature mortality, particularly from respiratory and cardiovascular diseases for humans. We have added descriptions about the pollutants reported to cause the health issues in the revised manuscript, see line 80-82.

Page 2 line 72: insert “in the area” after …piles exists”

We have added “in the area” in this sentence in the revised manuscript, see line 83-84.

Page 2 lines72-73: where are these 194 located?

The 194 sub-coal mines are located in the Yangquan coal mining district. We have added description of the location of these 194 sub-coal mines in the revised manuscript, see line 85.

Page 2 line 72: in the sentence ..”total amount of 1.0 gt”….amount of what?

Sorry for the mistake. Twenty one coal spoils piles exist in the area with a total amount of 1.0 Gt coal wastes. We have modified our statement in the revised manuscript, see line 84.

Describe Yangquan and how it is related or unrelated to the study site

The Yangquan coal mining district is located south-west of Yangquan city (Shanxi province, E113°36 N37°53). We have described Yangquan and its positional relation with the study site in the revised manuscript, see line 69-70.

Figure 1: which direction is north?

Sorry for our negligence. We have added the north direction in figure 2 in the revised manuscript, see line 108.

Page 3 lines 90-93: refer to figure

We have referred the figure (Figure 3) to the sentence in the revised manuscript, see line 115.

Page 3: Explain the purpose for collecting soil samples from around the water table

In this paper, we collect soil samples till the groundwater table to check the variation of PAHs contents in the vadose zone.

Experimental & analytical methods

Page 4 line 112: about 20 g of the Sample. Is this soil? Please indicate

These samples include coal spoils sample and soil samples. We have modified our statement in the revised manuscript, see line 136-137.

What was the purpose of hexamethylbenzene? Was it to check instrument sensitivity or recovery?

Yes, hexamethylbenzene is used as the internal standard to check the recovery. We have modified this sentence in the revised manuscript, see line 138-139.

Where the soil samples dried prior to PAH analysis? If so, please indicate

Selected coal spoils and soil samples were dried in a vacuum freeze dryer. We have added the place where the soil samples are dried in the revised manuscript, see line 134

Page 5 line 137-138: the recovery ratios for the surrogate.... reported ranges. What were these ranges?

We have added the recovery ratios for the surrogates in the samples in the revised manuscript, see line 170.

Results and discussion

Page 5 line 143: move to materials and methods

We have moved this sentence to the “materials and methods” section in the revised manuscript, see line 129-130

Page 5 lines 150 to 155: move to materials and methods

We have moved the statement to the “materials and methods” section in the revised manuscript, see line 143-149.

Provide standard deviations for all average concentrations.

We have added the standard deviations for all average concentrations in Table 1, Table 2 and Table S2 in the revised manuscript, see line 197-200, and 225-228.

Table S1 and S2 are the main data, please include these in the main paper and not as supplement.

We have moved the Tables S1 and S2 into the main paper as Table 1 and Table 2 in the revised manuscript, see line 197-200, and 225-228.

What is the possible reason for this trend and also explain the potential reasons why nap was most abundant PAH “The major compounds of PAHs in CSFGM is Nap (1556.8 ng/g), Acy (167.6 ng/g), 166 Ant (104.0 ng/g), Phe (84.60 ng/g), Flu (61.43 ng/g) and Fla (55.26 ng/g). Most of the 4-ring, 5-ring and 167 6-ring PAHs have lower contents in CSFGM compared to those in coal spoils.”

In general, low-ring PAHs (e.g., Nap) have a stronger migration behaviors than high-ring PAHs. Thus the major compounds of PAHs in CSFGM is Nap. We have modified our statement and explained the potential reasons why nap was the most abundant PAHs in the revised manuscript, see line 194-196.

Page 6 line 184: change “are” to “were”

We have changed “are” to “were” in the revised manuscript, see line 219.

Authors should explain why the low molecular weight PAHs were most abundant

We have added explanation why the low molecular weight PAHs were most abundant in the revised manuscript, see line 221-224.

Any reasons for these results? “Though the total PAHs content in S1 soil is not as high as S2 and S3, it is still higher than S4 and S5. Along the A-A’ profile a decreasing trend of PAHs in soil is observed.”

Along the A-A’ profile a decreasing trend of PAHs in soil is observed, predominantly due to the fact that total PAHs concentrations usually decrease with the increase of the closeness to a contamination source. We have added the reasons for the results “along the A-A’ profile a decreasing trend of PAHs in soil is observed” and the related reference for this in the revised manuscript, see line 235-237 and 470-471.

Page 7 lines 199, 200 and 202: replace “is” with “was”

We have replaced “is” with “was” in the revised manuscript, see line 241-245.

Page 7 line 206: provide distance for the bottom soil. Was it 2 m, 0.5 m or what?

It's 2 m. We have added the description of the distance for the bottom soil in the revised manuscript, see line 248.

Figures 5 & 6 have a very poor legibility.

We have redrawn Figures 5 & 6 in the revised manuscript, and now they have a good legibility, see line 254-257.

Page 7 line 218: replace “are” with “were”

We have replaced “are” with “were” in the revised manuscript, see line 258 and 260.

Page 7 line 220: did authors mean PAHs or PHAs?

Sorry for the mistake. We have corrected PAHs in the revised manuscript, see line 263.

Page 8 lines 225-226: Is there any reference to support the sentence “Based on the difference of adsorption ability of light and heavy PAHs on soil, the light PAHs is easier to move downward than heavy PAHs.”

We have added references to support the statement in the revised manuscript, see line 268 and 472-475.

Page 8 line 227: soil or top soil?

Yes, you are right. It should be top soil. We have modified this sentence in the revised manuscript, see line 273.

Page 8 lines 231-232: re-write the sentence “Therefore, as the major, even the only, 231 source of PAHs pollution in the coal spoils depositing area,”

We have modified our statement in the revised manuscript, see line 274-276.

Page 8 line 235: delete “test” after the heating

We have deleted “test” after “the heating” in the revised manuscript, see line 278.

Page 8 line 236: move “in the field” to after the “emission rate”.

We have moved “in the field” after “the emission rate” in the revised manuscript, see line 279.

Page 8 line 244: is 200 C considered low temperature? I also do not get the sentence because at high temperature combustion, the high molecular weight PAHs are abundant. Can authors explain this trend since lower molecular weight pahs contributed abundantly?

Previous studies indicated that there are three stages for the spontaneous combustion of coal spoils: low-temperature oxidation (<400 °C), spontaneous heating (600 °C) and spontaneous combustion stage (>800 °C). Therefore, in this study, 200 is considered to be low temperature. And we have modified our statement in the revised manuscript, see line 286.

At high temperature combustion, the low molecular weight PAHs are abundant. Therefore in this study, lower molecular weight Pahs contributed abundantly.

How different does heating or burning affect PAH production?

The heating and burning tests are employed to simulate the release of PAHs under different conditions. The heating test is under oxygen deficit condition while the burning test simulates the oxygen-enriched situation. The PAHs production in the heating test is about two times more than that in the burning test. And we have indicated this in the revised manuscript, see line 295.

Indicate p values for all statistical significance. Include SD for average concentrations.

We have indicated p values for all statistical significance and SD values for average concentrations in the revised manuscript, see Table 1, Table 2 and Table S2, see line 197-200, and 225-228.

Page 9 line 278: change “production of” to “products of”

We have changed “production of” to “products of” in the revised manuscript, see line 320.

Page 9 line 274: replace “are” with “were”

We have replaced “are” with “were” in the revised manuscript, see line 316.

Page 9 line 281: replace “compounds are” with “PAHs were”

We have modified this sentence in the revised manuscript, see line 323.

Page 10 line 282: assesse?

Sorry for the mistake. It should be “assesses”. We have changed “assesse” to “assess” in the revised manuscript, see line 324.

Page 9 lines 264-265: the highest ……followed by phe. Please delete

We have deleted these sentences in the revised manuscript, see line 323.

Reviewer 2 Report

This paper investigates PAH concentration and composition in coal spills, coal spoils fire gas mineral, leachate, and soils in a coal pile refuse sites at a coal mining area in northern China.  The overall objective is to evaluate potential environmental impacts.  The topic is important and the paper informative, but it is not well written to convincingly inform the reader as to environmental risk.  My main points are as follows.

The authors could do a better job of informing the reader as to potential risk.  One class of risk comes from spontaneous combustion from within the coal piles.  Many readers such as I may be unaware that this is a serious issue and this risk could be developed in more detail in the introduction.   The second risk seems to be from coal particles which can be transported some distance from the piles, dispersing risk over a broader area: I assume this dispersal comes from wind and water pathways.  The third risk is from infiltration into the soils.  For the study area, the lack of vegetation and the general disturbed nature of the landscape may facilitate the dispersal of coal particles over the greater landscape.  The nature of the soils, including coarse particles which may not be well compacted, would facilitate the downward movement of water and particles deeper into the soil.   Overall, the authors need to provide a clearer picture of risk pathways. Figure 2 is very instructive.

The methods seem adequate, but no information is provided on the number of replicates, if any.  Without replicates, the authors have a less convincing argument of spatial differences in PAH concentration and composition.  I assume that the coal spoils fire gas material refers to burned coal and this needs to be clarified.  Since the composition and concentration of PAHs in burned material is dependent on burn temperature and I assume this material is rather heterogenous, the authors could better inform the reader as to how they obtained a representative sample.  It looks like only one sample was collected. Also, it looks like only one leachate sample was collected.  It would have been a nice addition had they been able to measure ground water in the boreholes.  Figure 1 could also show where the samples were collected and is most useful.

More information could be given on the selection of temperatures for the combustion tests and how realistic they may be of the real world environment.

The results and discussion could be improved.  For example, for lines 150-167, a table would be easier to comprehend than the extended paragraph and figure 3.  The caption for figure 3 notes that the histograms are averages and so standard deviations or errors should be shown, ideally in a table.  Figure 3 could be retained either as a %composition figure and or a by ring number figure which would be more informative.  The same could be said for figure 4; redraw as a percent ring-composition figure with the concentrations in a table.  The histograms are had to read and the legends small. The scale for figure 6 also are hard to read.    

The authors really need to consider their graphs and how they can best display their points.  They are emphasising dominant composition and shifts in composition but these graphs do not show this well.   I would assume that CSFGM would have a different composition than coal spoils but this is hard to discern. Similarly, differences in PAH composition with distance from the piles and with depth are not easily discerned. I generally graph my individual PAH data as %composition histograms and show concentrations inside the histogram panel as text.  I think this would be more effective. Other people graph by ring number.

The authors have not got to the point of environmental risk as they do not have a section on interim sediment (and water) quality guidelines and probable effects levels.  Concentrations of individual PAHs should be investigated in this context.  I assume risk is to animals in the region and this could be expanded upon.

In summary, this paper has some potential but it does not really investigate environmental risk, information on replicates and variation around the mean or average are not given, and the figures do not convey well, the significances of differences in PAH composition between matrices.

Author Response

Dear Sir/Madam,

       Thank you so much for your kind review and valuable comments. The manuscript has been well prepared according to these comments. Great efforts were also made to correct the mistakes and improve the quality of the manuscript. The revisions in the manuscript are marked up highlighted in yellow color. Below is a point-by-point response.

Response to Reviewer 2 Comments

Open Review

English language and style

( ) Extensive editing of English language and style required 
(x) Moderate English changes required 
( ) English language and style are fine/minor spell check required 
( ) I don't feel qualified to judge about the English language and style 

Yes

Can be improved

Must be improved

Not applicable

Does the introduction provide sufficient background and   include all relevant references?

( )

( )

(x)

( )

Is the research design appropriate?

(x)

( )

( )

( )

Are the methods adequately described?

( )

(x)

( )

( )

Are the results clearly presented?

( )

( )

(x)

( )

Are the conclusions supported by the results?

( )

( )

(x)

( )

Comments and Suggestions for Authors

This paper investigates PAH concentration and composition in coal spills, coal spoils fire gas mineral, leachate, and soils in a coal pile refuse sites at a coal mining area in northern China.  The overall objective is to evaluate potential environmental impacts.  The topic is important and the paper informative, but it is not well written to convincingly inform the reader as to environmental risk. 

Thanks for your progressive suggestion. We have modified the paper and made great efforts to improve the quality of the manuscript. The revisions in the manuscript are marked up highlighted in yellow color.

My main points are as follows.

The authors could do a better job of informing the reader as to potential risk.  One class of risk comes from spontaneous combustion from within the coal piles.  Many readers such as I may be unaware that this is a serious issue and this risk could be developed in more detail in the introduction. The second risk seems to be from coal particles which can be transported some distance from the piles, dispersing risk over a broader area: I assume this dispersal comes from wind and water pathways.  The third risk is from infiltration into the soils.  For the study area, the lack of vegetation and the general disturbed nature of the landscape may facilitate the dispersal of coal particles over the greater landscape.  The nature of the soils, including coarse particles which may not be well compacted, would facilitate the downward movement of water and particles deeper into the soil.   Overall, the authors need to provide a clearer picture of risk pathways. Figure 2 is very instructive.

Thanks for your valuable review. Yes, you’re right. In a mining site and the surrounding areas, anthropogenic PAHs are raised from three sources: spontaneous combustion of coal spoils piles, transportation of coal particles from the piles and the infiltration of coal spoils into the soils. We have added the description of these three potential risks for the anthropogenic PAHs in the revised manuscript, see line 40-46.

The clear risk pathways of anthropogenic PAHs are provided in Figure 3 which can help us to better understand the environmental risks related to the PAHs issue.

The methods seem adequate, but no information is provided on the number of replicates, if any.  Without replicates, the authors have a less convincing argument of spatial differences in PAH concentration and composition.  I assume that the coal spoils fire gas material refers to burned coal and this needs to be clarified. 

The methods in this study are of three replications. We have added the information on the number of replicates in the revised manuscript, see line 167.

Thanks. The coal spoils fire gas material refers to the condensed gas materials from the burnt coal spoil piles which normally appeared in the top of the bank. For a detailed description, Pls reference to Querol, X. et al (2008): Environmental characterization of burnt coal gangue banks at Yangquan, Shanxi Province, China. Inter. J. Coal Geol. 2008, 75, 93-104.

Since the composition and concentration of PAHs in burned material is dependent on burn temperature and I assume this material is rather heterogenous, the authors could better inform the reader as to how they obtained a representative sample.  It looks like only one sample was collected. Also, it looks like only one leachate sample was collected.  It would have been a nice addition had they been able to measure ground water in the boreholes.  Figure 1 could also show where the samples were collected and is most useful.

Yes, you are right. The coal spoils, coal spoils fire gas mineral and soils are heterogenous. Therefore, representative samples were collected according to the sampling locations, sampling time, sampling randomicity and uniformity in this study. We have added the description of how we obtained a representative sample in the revised manuscript, see line 99-100.

Thanks for the progressive suggestion about the PAHs in groundwater in the boreholes. It gives us a good line of thinking. We will proceed it in our further studies to demonstrate how depth PAHs can infiltrate.

We have added the location of samples in Figure 2 in the revised manuscript, see line 108.

More information could be given on the selection of temperatures for the combustion tests and how realistic they may be of the real world environment.

Previous studies indicated that there are three stages for the spontaneous combustion of coal spoils: low-temperature oxidation (<400 °C), spontaneous heating (600 °C) and spontaneous combustion stage (>800 °C) [48]. In addition, the firing temperature for spontaneous combustion of coal spoils can be over 1000 °C [49-50]. Hence, in this study, coal spoils samples are heated/burnt in the sequence of 200 °C, 400 °C, 600 °C, 800 °C and 1000 °C for four hours. We have added the description for the realistic selection of temperatures for the combustion tests in the revised manuscript, see line 154-158.

The results and discussion could be improved.  For example, for lines 150-167, a table would be easier to comprehend than the extended paragraph and figure 3.  The caption for figure 3 notes that the histograms are averages and so standard deviations or errors should be shown, ideally in a table.  Figure 3 could be retained either as a %composition figure and or a by ring number figure which would be more informative.  The same could be said for figure 4; redraw as a percent ring-composition figure with the concentrations in a table.  The histograms are had to read and the legends small. The scale for figure 6 also are hard to read.

We have added a table (Table 1) in the revised manuscript, which makes the total PAHs and individual PAHs contents in coal spoils samples much more clear, see line 197-200.

We have added the standard deviations for all average concentrations in Table 1, Table 2 and Table S2 in the revised manuscript, see line 197-200, and 225-228.

We have redrawn Figure 3 and Figure 4 (now it is Figure 4 and Figure 5) as the percent ring-composition figures in the revised manuscript. In addition, the PAHs concentrations are illustrated in Table 1 and Table 2.

We have modified Figure 6 to make this figure clear for readers.

The authors really need to consider their graphs and how they can best display their points. They are emphasising dominant composition and shifts in composition but these graphs do not show this well.   I would assume that CSFGM would have a different composition than coal spoils but this is hard to discern. Similarly, differences in PAH composition with distance from the piles and with depth are not easily discerned. I generally graph my individual PAH data as %composition histograms and show concentrations inside the histogram panel as text.  I think this would be more effective. Other people graph by ring number.

Thank you so much for invaluable comments. We have redrawn Figure 3 and Figure 4 (now it is Figure 4 and Figure 5) and made them the percent ring-composition figures in the revised manuscript. Now they can better display the points.

The authors have not got to the point of environmental risk as they do not have a section on interim sediment (and water) quality guidelines and probable effects levels.  Concentrations of individual PAHs should be investigated in this context.  I assume risk is to animals in the region and this could be expanded upon.

Thank you so much for your comments. In this manuscript, we investigated the potential environmental impacts of PAHs on soil and water via spontaneous combustion and leaching of coal spoil piles through field investigation and indoor analysis. Studies on inter sediment and water quality guidelines and probable effects levels are really a good thought which can further our understanding on the environmental risks of PAHs, and we will consider it in further researches.

We have investigated the concentrations of individual PAHs in the revised manuscript, see line 186-193, 201-205, 217-222, 259-263, 271-272 and 289-293.

Yes, you are right. In this regions, impacts of PAHs on animals are reported. With a long-term exposure to PAHs, animals may suffer from an allergic skin response and breast tumors. We have added description of animals have been affected by PAHs in the revised manuscript, see line 82-83.

In summary, this paper has some potential but it does not really investigate environmental risk, information on replicates and variation around the mean or average are not given, and the figures do not convey well, the significances of differences in PAH composition between matrices.

Thanks so much for your valuable review comments. We have added information on the number of replicates in the revised manuscript, see line 167.

We have added the standard deviations for all average concentrations in Table 1, Table 2 and Table S2 in the revised manuscript, see line 197-200, and 225-228.

We have redrawn Figure 3 and Figure 4 and made them the percent ring-composition figures in the revised manuscript. Now they can better display the points.

In this manuscript, we investigated the potential environmental impacts of PAHs on soil and water via spontaneous combustion and leaching of coal spoil piles through field investigation and indoor analysis. Studies on differences in PAHs compositions between matrices/inter sediment and water quality guidelines are really a good thought which can further our understanding on the environmental risks of PAHs, and we will consider it in further researches. 

Round 2

Reviewer 1 Report

The manuscript has improved greatly after the response. 

Author Response

Response to Reviewer 1 Comments

Open Review

English language and style

( ) Extensive editing of English language and style required 
( ) Moderate English changes required 
(x) English language and style are fine/minor spell check required 
( ) I don't feel qualified to judge about the English language and style 

Yes

Can be improved

Must be improved

Not applicable

Does the introduction provide sufficient background and   include all relevant references?

(x)

( )

( )

( )

Is the research design appropriate?

(x)

( )

( )

( )

Are the methods adequately described?

(x)

( )

( )

( )

Are the results clearly presented?

(x)

( )

( )

( )

Are the conclusions supported by the results?

(x)

( )

( )

( )

Comments and Suggestions for Authors

The manuscript has improved greatly after the response. 

We have revised our manuscript based on each comment given by the reviewer. The authors would like to thank the anonymous reviewer for the helpful and constructive comments that greatly contributed to improving the quality of the paper.

Reviewer 2 Report

I have spent the good part of the day reviewing this paper and while it is improved, I still feel that it would benefit from another revision.  It is descriptive with many details mentioned but lacking specifics as to study sites, number of replicate samples, collected etc.  The main findings are buried in details and the paper is not a satisfactory read.   The fact that naphthalene and low molecular PAHs dominated the coal profiles needs to be noted and the implication that these volatile and water-soluble PAHs will be mobile in the environment through burning and leachate.  The higher molecular weight PAHs occurred as minor contributors to coal; it is surprisingly the concentrations were not that high and similar to concentrations that are considered background in northern Canadian environment. Therefore, the environmental risk seems low from these compounds including through combustion. Are the authors confident they are using the correct units for concentration?    Alternately, the sites are very sandy and this dilutes the coal particles and PAHs. There should be some reconciliation between the environmental damage reported and the PAH concentrations. Possibly damage is associated with metals or lower pHs caused by sulfur in the coal.  Specific points.

Line 59.  What are gobs?

Line 98. How many samples were collected?  Figure 2 shows a collection zone between 2 (a coal mine) and 15 (a coal pile).  Why were samples collected only at one coal pile?  The figure is very busy showing information that is not relevant to the paper, i.e., they sampled only coal pile site out of 21 sites.  Did the authors collect replicate samples or did they perform replicate analyses on a single sample (line 163)?

Line 191. Why is ACY so much higher in CFSGM that coal spills – more than 55 times greater?

Lines 193-194.  The text indicates that three replicate samples were analyzed but the table suggests that four coal spoil samples were collected.  If the authors collected three replicate analyses per sample, they should show the standard deviations.  The CSFGM data could appear below the coal spoil data and the average and standard deviation for improved clarity: a space could separate the two data sets. Again, it looks like a single sample was collected.  Abbreviations should be spelled out to save reader looking for explanation in the text.  

For Table 1, the authors could compare concentrations to CCME sediment quality guidelines for PAHs. This information is accessible at http://st-ts.ccme.ca/en/index.html.  Naphthalene guidelines are exceeded for all samples and Acy, Ace, Flu, Phe and Ant for CSFGM. While these guidelines are for aquatic life and not soils and terrestrial life, the authors could push it and suggest that if these coal particles were deposited on a lake or river, lake sediments would be below sediment quality guidelines for most PAHs.  

The very strong dominance of naphthalene in the coal spoils is unexpected.  Moreover, the percent composition is about the same as for CSFGM when one would expect it to be lower given that naphthalene is volatile and should burn off.  Similarly, naphthalene is water-soluble and one would expect a higher percentage in the leachate than the coal spoils.   I did do some searching for PAH composition of coals and found a reference that shows that lignite A is high in naphthalene so some discussion around this should discuss the fact that that some coals are high in naphthalene; see link below which has some nice graphs.  The primary reference for the link below is Stout and Emsbo-Mattingly 2008. Concentration and character of PAHs and other hydrocarbons in coals of varying rank – Implications for environmental studies of soils and sediments containing particulate coal.  Organic Chemistry 39:801-809.  However, this is not a free download and I had to access through a university library account I have.   

https://books.google.ca/books?id=YgkuwfizPd0C&pg=PA187&lpg=PA187&dq=coal+pah+profile&source=bl&ots=PhS0qtazao&sig=ACfU3U2_G0OHFC_duDePPbc_GewWx6vR4w&hl=en&sa=X&ved=2ahUKEwii_8ro4fLgAhUPlKwKHZCpCeMQ6AEwD3oECAEQAQ#v=onepage&q=coal%20pah%20profile&f=false

Line 197. Are the leachate data based on a single sample?

Figure 4 and others of this type could be redrawn if the author’s graphic packages allow this.  That is, they could have two Y-axes with the left axis for naphthalene and the current scale and the right for minor constituents and the scale going up to 10%.  This would show the composition of minor PAHs better.

Table 2. The borehole sampling sites are shown in figure 2 but table 2 should include a column for depth.  The soil sampling sites are not shown on figure 2 but are not numbered.  Averages and standard deviations are shown for soil samples but not bore hole samples.  What does “all these statistical data at the 0.05 level mean? For soil samples, PAH concentration may be a function of soil composition with respect to grain size and organic carbon.  So without these measurements and replicate samples at each site, hard to infer much about spatial differences. The data almost could be combined to give a mean and standard deviation.  Figure 5 and (fig 4) should have error bars for each histogram, presuming that replicates were collected and/or analyses done.

Figures 6 and 7 repeat data in table 2 and are not needed.  They could be redrawn as a cumulative percent composition graph with depth.  

Line 267.  The leachate data should be shown in the main body of the paper and again as composition although in rereading the paper I see it is referred to in figure 4. This figure is not mentioned on line 267. None of the PAHs exceeded CCME guidelines for the protection of aquatic life. Lines 270-272 make no sense.

Lines 290-301 could be improved. Without error bars, cannot say that differences in PAH release differ with temperature. It makes no sense that less PAHs would be released at higher temperatures. Figures 8 and 9 have the same X and Y-axes and legend but different data so something is wrong here

Lines 314-316.  The findings are overstated.  No information is provided on rate of combustion and deposition in the study area.

In summary, this paper still needs major work.  It is highly descriptive, lacks a number of important details including sample locations, the number of replicate samples collected per location, and the number of replicate analyses performed per sample.   The high concentrations and percent naphthalene is of interest and the authors could make more of this and how this relates to the specific type of coal that is at this mine site.  Moreover, they could build the paper around this including noting that naphthalene remains the major product in CSFGM (which seems peculiar to me) and in leachate.  The lower percent composition in leachate seems strange given the high water solubility of naphthalene and requires explanation. The major threat to the environment appears to come from the low molecular weight PAHs, some of which exceed CCME guidelines, which, if coal particles and dust were deposited onto aquatic landscape could pose some risk.  Less risk seems to come from the higher molecular weight PAHs.  Overall, apart from naphthalene and possibly Acy and Ace, the PAH concentrations are not high and comparable to concentrations observed in non-impacted regions in northern Canada. A broader comparison of their results with studies around other coal sites (including coal-fired power plants) would be helpful.    

Author Response

Response to Reviewer 2 Comments

Open Review

English language and style

( ) Extensive editing of English language and style required 
(x) Moderate English changes required 
( ) English language and style are fine/minor spell check required 
( ) I don't feel qualified to judge about the English language and style 

Yes

Can be improved

Must be improved

Not applicable

Does the introduction provide sufficient background and   include all relevant references?

( )

(x)

( )

( )

Is the research design appropriate?

( )

(x)

( )

( )

Are the methods adequately described?

( )

(x)

( )

( )

Are the results clearly presented?

( )

( )

(x)

( )

Are the conclusions supported by the results?

( )

( )

(x)

( )

Comments and Suggestions for Authors

I have spent the good part of the day reviewing this paper and while it is improved, I still feel that it would benefit from another revision. It is descriptive with many details mentioned but lacking specifics as to study sites, number of replicate samples, collected etc. The main findings are buried in details and the paper is not a satisfactory read.   The fact that naphthalene and low molecular PAHs dominated the coal profiles needs to be noted and the implication that these volatile and water-soluble PAHs will be mobile in the environment through burning and leachate. The higher molecular weight PAHs occurred as minor contributors to coal; it is surprisingly the concentrations were not that high and similar to concentrations that are considered background in northern Canadian environment. Therefore, the environmental risk seems low from these compounds including through combustion. Are the authors confident they are using the correct units for concentration?  Alternately, the sites are very sandy and this dilutes the coal particles and PAHs. There should be some reconciliation between the environmental damage reported and the PAH concentrations. Possibly damage is associated with metals or lower pHs caused by sulfur in the coal.  Specific points.

The authors would like to thank the anonymous reviewer for the helpful and constructive comments that greatly contributed to improving the quality of the paper.

We have revised our manuscript based on each comment given by the reviewer. The revisions in the manuscript are marked up highlighted in yellow color.

We have added details about the study sites, replicate samples and sample collection in the revised manuscript, see line 69-71, 94-101, and 158.

Thanks so much for your invaluable suggestion. We have added the statement about the naphthalene and low molecular PAHs dominated the coal profiles and the reasons in the revised manuscript, see line 189-192.

We have checked the data and units for PAHs concentration carefully. And we are sure that the data and the units are correct. It is worthwhile to note that the PAHs in this study area are approximately 1.8 ~ 91 times higher than those reported case studies for coal gangue samples in Hong Kong (170 ng/g) (Zhang et al., 2006) and Henan province (347 ng/g) (Wang et al., 2009) in China and Serrinha (6.67 ng/g) (Ribeiro et al., 2012).

Much thanks for your advice. Studies about the possibly damage of PAHs associated with metals or lower pH is really a good thought which can further our understanding of the potential environmental concerns of PAHs in the coal, and we will consider it in further researches.

Line 59.  What are gobs?

We have replaced gobs with piles in the revised manuscript, see line 59.

Line 98. How many samples were collected?  Figure 2 shows a collection zone between 2 (a coal mine) and 15 (a coal pile).  Why were samples collected only at one coal pile?  The figure is very busy showing information that is not relevant to the paper, i.e., they sampled only coal pile site out of 21 sites.  Did the authors collect replicate samples or did they perform replicate analyses on a single sample (line 163)?

We have added the description of the numbers of samples collected in the study area in the revised manuscript, see line 94-97. In this study, 23 samples were collected, including one coal spoils fire gas mineral sample (CSFGM), four coal spoils, five soil samples in the profile of A-A’, seven soil samples from bore hole B1, five soil samples from bore hole B2 and one leachate discharge sample from the coal spoils piles.

The CSFGM with yellow color is distributed dispersedly (Figure 2). We collected them by separating them from the soil. So the sample is a mixture. Additionally, the coal spoils were deposited like a small hill along the slope of the mountain. So the infiltration water is merged together into the valley and discharged like a spring in the foot of the bank. The leachate sample collected is the representative of the whole bank.

Yes, we only collected samples from coal spoils banks No. 15 (coal mine No. 2). This is because coal spoils banks No.15 is the hugest coal spoils banks with coal waste from the mining activity of the major coal seams, and it is very representative.

Thank you for your advice. We have removed Figure 2 to the Supplementary Information as Figure S1.

We have modified our statement in the revised manuscript, see line 158. In this study, we perform three replicate analyses per sample.

Line 191. Why is ACY so much higher in CFSGM that coal spills – more than 55 times greater?

We have added explanation for the higher PAHs in CFSGM than that in coal spoils in the revised manuscript, see line 187-192. PAHs in CFSGM is higher than that in coal spoils, this is because during the incomplete combustion of coal spoil, large amounts of PAHs are released into the atmosphere, thereby entering into CSFGM because of its hydrophobic nature (Pone et al., 2007). The major compounds of PAHs in CSFGM was Nap (1556.8 ng/g), Acy (167.6 ng/g), Ant (104.0 ng/g), Phe (84.60 ng/g), Flu (61.43 ng/g) and Fla (55.26 ng/g), due to the strong migration behaviors for these low-ring PAHs, like Nap and Acy.

Lines 193-194.  The text indicates that three replicate samples were analyzed but the table suggests that four coal spoil samples were collected.  If the authors collected three replicate analyses per sample, they should show the standard deviations.  The CSFGM data could appear below the coal spoil data and the average and standard deviation for improved clarity: a space could separate the two data sets. Again, it looks like a single sample was collected.  Abbreviations should be spelled out to save reader looking for explanation in the text.  

In this study, we performed three replicate analyses per sample. We have added the standard deviations for coal spoil samples and modified Table 1 in the revised manuscript, see line 199-202. Now the CSFGM data and the coal spoil data are much more clear.

As shown in Figure 2, the CSFGM with yellow color is distributed dispersedly. We collected them by separating them from the soil. So the sample is a mixture.

We have spelled out the abbreviations in Table 1 in the revised manuscript, see line 199-200.

For Table 1, the authors could compare concentrations to CCME sediment quality guidelines for PAHs. This information is accessible at http://st-ts.ccme.ca/en/index.html.  Naphthalene guidelines are exceeded for all samples and Acy, Ace, Flu, Phe and Ant for CSFGM. While these guidelines are for aquatic life and not soils and terrestrial life, the authors could push it and suggest that if these coal particles were deposited on a lake or river, lake sediments would be below sediment quality guidelines for most PAHs.  

Thanks so much for your progressive suggestion. We have added the comparison for PAHs between the samples collected in this study area and CCME sediment quality guidelines in the revised manuscript, see line 193-198 and Table 1.

The very strong dominance of naphthalene in the coal spoils is unexpected.  Moreover, the percent composition is about the same as for CSFGM when one would expect it to be lower given that naphthalene is volatile and should burn off.  Similarly, naphthalene is water-soluble and one would expect a higher percentage in the leachate than the coal spoils.   I did do some searching for PAH composition of coals and found a reference that shows that lignite A is high in naphthalene so some discussion around this should discuss the fact that that some coals are high in naphthalene; see link below which has some nice graphs.  The primary reference for the link below is Stout and Emsbo-Mattingly 2008. Concentration and character of PAHs and other hydrocarbons in coals of varying rank – Implications for environmental studies of soils and sediments containing particulate coal.  Organic Chemistry 39:801-809.  However, this is not a free download and I had to access through a university library account I have.

 https://books.google.ca/books?id=YgkuwfizPd0C&pg=PA187&lpg=PA187&dq=coal+pah+profile&source=bl&ots=PhS0qtazao&sig=ACfU3U2_G0OHFC_duDePPbc_GewWx6vR4w&hl=en&sa=X&ved=2ahUKEwii_8ro4fLgAhUPlKwKHZCpCeMQ6AEwD3oECAEQAQ#v=onepage&q=coal%20pah%20profile&f=false

Thank you so much for the impressive suggestion. We have added discussion about the very strong dominance of naphthalene in the coal spoils in the revised manuscript, see line 182-184 and 480-483.

Line 197. Are the leachate data based on a single sample?

Yes, we collected 1 leachate sample at the front foot of the coal spoils bank. The coal spoils were deposited like a small hill along the slope of the mountain. So the infiltration water is merged together into the valley and discharged like a spring in the foot of the bank. The leachate sample collected is the representative of the whole bank. Additionally, we did three replicate analyses for this sample.

Figure 4 and others of this type could be redrawn if the author’s graphic packages allow this.  That is, they could have two Y-axes with the left axis for naphthalene and the current scale and the right for minor constituents and the scale going up to 10%.  This would show the composition of minor PAHs better.

Thank you for your advice. We have redrawn Figure 4 and Figure 5 (now they are Figure 3 and Figure 4) in the revised manuscript, and now it shows the compositions of constituents more clear, see line 213-216 and 238-240.

Table 2. The borehole sampling sites are shown in figure 2 but table 2 should include a column for depth.  The soil sampling sites are not shown on figure 2 but are not numbered.  Averages and standard deviations are shown for soil samples but not bore hole samples.  What does “all these statistical data at the 0.05 level mean? For soil samples, PAH concentration may be a function of soil composition with respect to grain size and organic carbon.  So without these measurements and replicate samples at each site, hard to infer much about spatial differences. The data almost could be combined to give a mean and standard deviation.  Figure 5 and (fig 4) should have error bars for each histogram, presuming that replicates were collected and/or analyses done.

We have inserted a column for depth in Table 2 (now it is Table 3) in the revised manuscript, see line 231-237.

We have numbered the soil samples (S1-S5) in Figure 2 in the revised manuscript, see line 111-118.

We have added the averages and standard deviations for the bore hole samples in Table 3 in the revised manuscript, see line 231-237.

We have removed the sentence “all these statistical data at the 0.05 level” in the revised manuscript, see line 237.

We have added the averages and standard deviations for soil samples in Table 3 in the revised manuscript, see line 231-237. We did not do the relation between PAH concentration and soil composition (with respect to grain size and organic carbon), but it is really a good thought, and we will consider it in further researches.

We have modified Figure 3 and Figure 4 by adding error bars in the revised manuscript, see line 213-216 and 238-240.

Figures 6 and 7 repeat data in table 2 and are not needed.  They could be redrawn as a cumulative percent composition graph with depth.  

We have redrawn Figure 6 and 7 (now they Figure 5 and 6) as cumulative percent composition graphs with depth in the revised manuscript, see line 264-266 and 286-288.

Line 267.  The leachate data should be shown in the main body of the paper and again as composition although in rereading the paper I see it is referred to in figure 4. This figure is not mentioned on line 267. None of the PAHs exceeded CCME guidelines for the protection of aquatic life. Lines 270-272 make no sense.

We have moved the composition of the leachate sample into the main body as Table 2 in the revised manuscript, see line 210-212.

We have mentioned Figure 3 and Table 2 in the revised manuscript, see line 280.

The PAHs in the leachate sample do not exceed CCME guidelines. Generally, PAHs concentration is high in leachate samples due to the weathering dissolution of newly coal spoils. The leachate sample we collected was normally deposited longer than two years. The reason we collected this sample is that we want to carry out comparison for PAHs between leachate sample for newly coal spoils and for deposited longer time.

Lines 290-301 could be improved. Without error bars, cannot say that differences in PAH release differ with temperature. It makes no sense that less PAHs would be released at higher temperatures. Figures 8 and 9 have the same X and Y-axes and legend but different data so something is wrong here

Temperature is one of the main factors affecting the PAH emissions from coal spoils combustion. Plenty of researches show that during the coal spoils combustion, the amount of each PAH emitted may be variable. PAHs concentration released increase at first and then descend with the increasing of temperature. The peak data for PAHs occurs at 700-800 (Mastral et al., 1996, 1999; Tobias et al., 1995; Liu et al., 2000; Yan et al., 2002). Thus as the temperature goes up to 1000 , less PAHs would be released.

Figure 8 and Figure 9 (now they are Figure 7 and Figure 8) show the release of PAHs in heating test and in burning test at different temperature, respectively. Therefore, they have the same X and Y-axes and legend but different data.

Lines 314-316.  The findings are overstated.  No information is provided on rate of combustion and deposition in the study area.

We have modified the statement in the revised manuscript, see line 329-334.

In summary, this paper still needs major work.  It is highly descriptive, lacks a number of important details including sample locations, the number of replicate samples collected per location, and the number of replicate analyses performed per sample.   The high concentrations and percent naphthalene is of interest and the authors could make more of this and how this relates to the specific type of coal that is at this mine site.  Moreover, they could build the paper around this including noting that naphthalene remains the major product in CSFGM (which seems peculiar to me) and in leachate.  The lower percent composition in leachate seems strange given the high water solubility of naphthalene and requires explanation. The major threat to the environment appears to come from the low molecular weight PAHs, some of which exceed CCME guidelines, which, if coal particles and dust were deposited onto aquatic landscape could pose some risk.  Less risk seems to come from the higher molecular weight PAHs.  Overall, apart from naphthalene and possibly Acy and Ace, the PAH concentrations are not high and comparable to concentrations observed in non-impacted regions in northern Canada. A broader comparison of their results with studies around other coal sites (including coal-fired power plants) would be helpful.    

We have added details about the study sites, replicate samples and sample collection in the revised manuscript, see line 69-71, 94-101, and 158.

Thanks so much for your invaluable suggestion. We have added more details about the high concentrations and percent of Naphthalene in the revised manuscript, see line 182-184 and 187-192. Studies about how this relates to the specific type of coal is a good thought and a big project, which we can proceed in further researches.

The PAHs in the leachate sample are in lower percent composition although naphthalene is highly water-soluble. This is because that the water is not the major medium for top soil PAHs pollution. Coal spoils constitute the major and even the only source of PAHs pollution in the coal spoils depositing area.

Yes, you’re right. We have added the comparison for PAHs between the samples collected in this study area and CCME sediment quality guidelines in the revised manuscript, see line 193-198 and Table 1.

Round 3

Reviewer 2 Report

This paper is much improved and I appreciate the authors’ taking my comments to heart in improving the paper.  However, I still have some lingering issues. 

The English should be improved and I hope an editor will deal with these matters.  For examples lines 16-17 on “indoor analyses” needs to be rewritten.  Line 35 “widely” should be “wide”.  Line 36: PAHs are abundant in the “universe” probably could be changed to “earth”

For figure 1, it is not clear to me where B1 and B2 are on the photograph and if S1-S5 are along an elevation gradient. Please clarify.

The experimental methods for the combustion tests are unclear to me.  I assume a homogenous sample of coal spills was obtained and then divided into subsamples, which were then weighed.  Were replicate experiments conducted with separate samples for each of the five temperatures and the two burn/heat conditions, i.e., were 10 samples in total run or 30 if there were three replicates? Lines 156-157 suggest measurements. I assume that the amount of PAH captured in resin column was calculated as a weight and then that weight divided by the original weight of each sample.  Data could also be expressed as a rate with time brought into the calculations. Were the samples weighed after the experiment, which would allow for an estimate of total loss of material and more effectively comparisons of spoil versus CSFGM samples, i.e, higher PAH concentrations in later related to a concentration effect with burning.

I assume it took about an hour for the 1000C samples to reach that temperature at the heating rate given in the text.

Lines 188-198 are confusing. I still do not understand why PAH concentrations are higher burned samples unless there is a weight loss of elements such as carbonates.  Also is it possible that the samples are somewhat more porous or in a region where they were strongly impacted by leachate?

Table 2 requires a row caption on top of table.

Figure 5 and 6 captions – they should be “concentration” and not “release”.

Figures 7 and 8. I still do not understand why less PAH is released under oxygenated conditions.  The captions should incorporate that heating is a low oxygen simulation for deeper in the pile and burning an oxygen simulation at the pile surface.  Unless a time variable is brought in, the measurement is concentration and not release which is a rate. Error bars should be shown for the total concentration.  This part needs tightening up.

To conclude, an improved paper but still requires editing for improved clarity, English and better explanation.

Author Response

Reply to reviewers:

Really thanks for your and the anonymous reviewers’ advice. Your helpful and constructive comments greatly contribute to improving the quality of the manuscript. Please find my point-by-point revision based on the reviewers’ comments and the revised manuscript.

We have revised our manuscript based on each comment given by the reviewers. You can find all of the changes highlighted in the manuscript with change mark. I hope that the revisions and responses to all of the comments will result in the timely publication of this work.

Comments and Suggestions for Authors

This paper is much improved and I appreciate the authors’ taking my comments to heart in improving the paper.  However, I still have some lingering issues. 

The authors would like to thank the anonymous reviewer for the helpful and constructive comments that greatly contributed to improving the quality of the paper.

We have revised our manuscript based on each comment given by the reviewer. The revisions in the manuscript are marked up highlighted in yellow color.

The English should be improved and I hope an editor will deal with these matters.  For examples lines 16-17 on “indoor analyses” needs to be rewritten.  Line 35 “widely” should be “wide”.  Line 36: PAHs are abundant in the “universe” probably could be changed to “earth”.

Thanks so much for your invaluable suggestions. The manuscript has been edited by an English-speaking native to improve the English.

We have modified our statement in the revised manuscript, see line 16-17.

We have changed “widely” to “wide” in the revised manuscript, see line 35.

We have replaced “universe” with “earth” in the revised manuscript, see line 36.

For figure 1, it is not clear to me where B1 and B2 are on the photograph and if S1-S5 are along an elevation gradient. Please clarify.

We have redrawn Figure 1 (now it is Figure 2) in the revised manuscript, see line 111. Now it shows the locations of B1 and B2 clearly.

We have modified our statement in the revised manuscript, see line 100-101. The surface soil samples were collected in the front of the coal spoils banks, with a distance of 5, 10, 50, 100 and 500 m. Therefore, S1-S5 samples are not along an elevation gradient.

The experimental methods for the combustion tests are unclear to me.  I assume a homogenous sample of coal spills was obtained and then divided into subsamples, which were then weighed. 

Yes, the coal spoils samples were collected, and crushed to powder. Then the powder are mixed homogeneously and dividied into subsamples for combustion tests.

Were replicate experiments conducted with separate samples for each of the five temperatures and the two burn/heat conditions, i.e., were 10 samples in total run or 30 if there were three replicates?

In this study, there are three replicates for each sample. Breifly, each sample was conducted for the five temperatures and the two burn/heat conditions and repeated three times.

Lines 156-157 suggest measurements. I assume that the amount of PAH captured in resin column was calculated as a weight and then that weight divided by the original weight of each sample.  Data could also be expressed as a rate with time brought into the calculations. Were the samples weighed after the experiment, which would allow for an estimate of total loss of material and more effectively comparisons of spoil versus CSFGM samples, i.e, higher PAH concentrations in later related to a concentration effect with burning.

Yes, the PAHs in the resin was calculated as a weight (ng/g), and then that weight divided by the original PAHs content in the samples.

Much thanks for your advice. Studies about the effectively comparisons of coal spoils versus CSFGM samples are a really good thought and we will consider it in our further researches.

I assume it took about an hour for the 1000C samples to reach that temperature at the heating rate given in the text.

Yes. The heating experiments are conducted in a horizontal tubular reactor by using resistance wire heating. In this study, coal spoils samples were heated in the sequence of 200°C, 400 °C, 600 °C, 800 °C and 1000 °C and kept stable for four hours. The temperature increased at a rate of 20°C/min. Therefore, it took about 50-60 minutes for the heating temperature to reach 1000 °C, and the coal spoils samples were kept stable in 1000 °C for another 3 hours.

Lines 188-198 are confusing. I still do not understand why PAH concentrations are higher burned samples unless there is a weight loss of elements such as carbonates.  Also is it possible that the samples are somewhat more porous or in a region where they were strongly impacted by leachate?

PAHs are released into the atmosphere in the gaseous form. And then, it condensed down together with sulfur. Therefore, higher contents of PAHs in coal spoil fire gas mineral (CSFGM) sample are observed than those in coal spoil samples.

Generally, there are interspace or fractures in the coal gangue hills. Coal spoils may be strongly impacted by leachate when there are precipitation.

Table 2 requires a row caption on top of table.

Thank you so much for your invaluable suggestion. We have added a row caption on top of table in Table 2 in the revised manuscript, see line 210-211.

Figure 5 and 6 captions – they should be “concentration” and not “release”.

We have replaced “release” with “concentration” in the Figures 5 and 6 in the revised manuscript, see line 265-266 and 286-287.

Figures 7 and 8. I still do not understand why less PAH is released under oxygenated conditions.  The captions should incorporate that heating is a low oxygen simulation for deeper in the pile and burning an oxygen simulation at the pile surface.  Unless a time variable is brought in, the measurement is concentration and not release which is a rate. Error bars should be shown for the total concentration.  This part needs tightening up.

Under oxygenated conditions, the hydrocarbons in coal spoils normally react with the oxygen during the combustion process to form water vapor (H2O) and carbon dioxide (CO2). Less PAHs are released. Whilst under oxygen deficit condition, coal spoils burn incompletely, during which more PAHs are yielded and released. Therefore, the burning test (oxygen-enriched situation) produces a lower content of PAHs than that in heating test (oxygen deficit condition).

We have modified the captions of Figures 7 and 8 in the revised manuscript, see line 322-325.

We have replaced “emission rate” with “emission” in the manuscript, see line 289-290. In this manuscript, emission rate represents the percentage of PAHs released from coal spoils (weight percentage).

We have added error bars for the total PAHs concentrations in the burning and heating test in the revised manuscript, see line 296 and 305-307.

To conclude, an improved paper but still requires editing for improved clarity, English and better explanation.

We have modified our manuscript based on each comment given by the reviewer. Great efforts are made to improve the English of the manuscriptsee line 14-17, 25, 31, 35-36, 68, 76, 81, 88, 114, 123, 182-183, 191, 194, 208, 218, 247, 254, 277, 318, 335. The authors would like to thank the anonymous reviewer for the helpful and constructive comments that greatly contributed to improving the quality of the paper.
